Protist taxonomic and functional diversity in aquatic ecosystems of the Brazilian Atlantic Forest

Carvalho da Silva Vanessa
http://orcid.org/0000-0002-3332-1869 Fernandes Noemi noemi.mfernandes@gmail.com
Instituto de Recursos Naturais, Programa de Pós-Graduação em Meio Ambiente e Recursos Hídricos, Universidade Federal de Itajubá , Itajubá, Minas Gerais , Brazil
Franco Bernardo
Electronic publication date: 2023 Aug 1
Publication date: 2023
Volume: 11
Electronic Location ID: e15762
Received 2023 Apr 17; Accepted 2023 Jun 27
Copyright year: 2023
License: This is an open access article, free of all copyright, made available under the Creative Commons Public Domain Dedication. This work may be freely reproduced, distributed, transmitted, modified, built upon, or otherwise used by anyone for any lawful purpose.
License URL: https://creativecommons.org/publicdomain/zero/1.0/

Keywords: Coastal lagoons, DNA metabarcoding, Eukaryotic diversity, Inland ecosystems, Neotropics, South America, Protist diversity

Funding: Fundação Carlos Chagas Filho de Amparo à Pesquisa do Estado do Rio de Janeiro E-202.325/2018 This work was supported by the Fundação Carlos Chagas Filho de Amparo à Pesquisa do Estado do Rio de Janeiro (FAPERJ, No. E-202.325/2018). The funders had no role in study design, data collection and analysis, decision to publish, or preparation of the manuscript.

==============================
The Brazilian Atlantic Forest and its associated ecosystems are highly biodiverse but still understudied, especially with respect to eukaryotic microbes. Protists represent the largest proportion of eukaryotic diversity and play important roles in nutrient cycling and maintenance of the ecosystems in which they occur. However, much of protist diversity remains unknown, particularly in the Neotropics. Understanding the taxonomic and functional diversity of these organisms is urgently needed, not only to fill this gap in our knowledge, but also to enable the development of public policies for biological conservation. This is the first study to investigate the taxonomic and trophic diversity of the major protist groups in freshwater systems and brackish coastal lagoons located in fragments of the Brazilian Atlantic Forest by DNA metabarcoding, using high-throughput sequencing of the gene coding for the V4 region of the 18S rRNA gene. We compared α and β diversity for all protist communities and assessed the relative abundance of phototrophic, consumer, and parasitic taxa. We found that the protist communities of coastal lagoons are as diverse as the freshwater systems studied in terms of α diversity, although differed significantly in terms of taxonomic composition. Our results still showed a notable functional homogeneity between the trophic groups in freshwater environments. Beta diversity was higher among freshwater samples, suggesting a greater level of heterogeneity within this group of samples concerning the composition and abundance of OTUs.Ciliophora was the most represented group in freshwater, while Diatomea dominated diversity in coastal lagoons.

Introduction

It is widely known that microorganisms dominate the diversity on Earth and the protists, a paraphyletic assemblage of single-celled organisms, represent a significant part of this diversity (Adl et al., 2019; Burki et al., 2020). Protists can be found in a variety of habitats, often representing the largest portion of eukaryotic richness (de Vargas et al., 2015; Mahé et al., 2017; Singer et al., 2019; Obiol et al., 2020). Although they are very common, present in virtually all environments, molecular surveys of biodiversity has revealed that most of the taxonomic diversity of protists remains undescribed (Bass & Boenigk, 2011; Pawlowski et al., 2012; del Campo et al., 2014). This is especially evident in less explored regions such as the Neotropics (Dunthorn et al., 2012; Lentendu et al., 2019; Fernandes et al., 2021; Ritter et al., 2021; Câmara et al., 2022). This gap in knowledge about the taxonomic and functional diversity of protists is an obstacle to a clearer view of how ecosystems operate (Sherr et al., 2007; López-García & Moreira, 2008).

Protists are key components in the ecosystems they inhabit. They can be found as free-living forms, but many species are symbionts (i.e., parasites, parasitoids, mutualists and commensals) on a wide range of hosts, including other protists, plants and metazoans, directly affecting the ecological aspects and controlling populations of their hosts (Chambouvet et al., 2008; Nowack & Melkonian, 2010; Edgcomb, 2016). These parasitic and mutualistic symbionts can dominate the diversity and abundance in several environments (Guillou et al., 2008; Geisen et al., 2015; Mahé et al., 2017). In addition to being a source of food for many organisms, protists act at the base of food chains as primary producers, as consumers of bacteria or other protists, and as decomposers (Wetzel, 2001; Corliss, 2002). There are also protists capable of performing both photosynthesis and phagotrophy, according to their life cycle stage or environmental conditions, in a type of nutrition called mixotrophy (Jones, 2001; Mitra et al., 2014). Despite the importance of understanding the functional profile of protist communities in different environments (Singer et al., 2021), this type of investigation has rarely been done in Brazilian ecosystems (de Araujo et al., 2018).

The constant variation of the physico-chemical conditions and an overlap with the microbial communities from the adjacent soil could favor the development of a highly diverse and dynamic protistan assemblage in freshwater systems (Debroas et al., 2017; Boenigk et al., 2018). However, molecular studies have shown that transitional environments, such as brackish coastal lagoons and estuaries, also have high protist diversity (Schubert et al., 2011; Telesh et al., 2011; Telesh, Schubert & Skarlato, 2013; Grinienė et al., 2019), contrary to what was previously believed (Remane, 1934).

High-throughput sequencing of molecular markers from environmental samples, known as metabarcoding, are powerful tools to describe the diversity of protists (de Vargas et al., 2015) and have expanded our knowledge about the phylogenetic placement of these organisms and uncovered a high number of new lineages (Jamy et al., 2020; Rajter & Dunthorn, 2021; Czech et al., 2022). In under-explored regions with high potential for discovering new taxa, such as the Brazilian biomes (Fernandes et al., 2021; Câmara et al., 2022), this tool is even more promising.

The Atlantic Forest is one of the top two Brazilian ecosystems richest in plant and animal diversity and endemism (Mittermeier et al., 1999) and the world’s fourth leading biodiversity hotspot (Myers et al., 2000). At the same time, this is one of the global most depleted habitats, retaining only a small part of its primary vegetation (Mittermeier et al., 1999). A number of associated habitats such as mangroves, rivers, streams, creeks, lakes, and lagoons are included in this biome. More than 90% of the Atlantic Forest is within the Brazilian territory, therefore, its conservation is largely a Brazilian concern (Marques & Grelle, 2021). While the diversity of plants and vertebrates is relatively well documented, little is known about its microbial diversity (Pontes, 2015; Ritter et al., 2021). To the best of our knowledge, only two article have been published so far on the molecular diversity of protists in the Brazilian Atlantic Forest through DNA metabarcoding and both dealt exclusively with the diversity of the phylum Ciliophora (Simão et al., 2017; Fernandes et al., 2021). This is the first study to examine the taxonomic and trophic diversity of the major protist groups in water bodies located in the Atlantic Forest by DNA metabarcoding. We compared the α and β diversity among samples for the overall protists communities and assessed the relative abundance of phototrophic, consumers, and parasitic taxa in brackish coastal lagoons and freshwater systems, also contributing to a better understanding of the dynamics and adaptations of protists to different salinity levels.

Materials and Methods

Sampling

Samples of freshwater and brackish water were obtained from 23 sites located in fragments of the Atlantic Forest in Rio de Janeiro state, Brazil (Fig. 1), as detailed in Fernandes et al. (2021). Five aliquots of 200 mL of water and resuspended sediment were collected along the edges of each sampling site, making up a total volume of 1 L per sample. The samples were stored in sterile plastic containers and then taken to the laboratory for filtration and DNA extraction less than 24 h after sampling. The total volume was filtered with a peristaltic pump through 0.22 µm Polyethersulfone (PES) membranes (75 mm diameter) and the retained content (about 0.5 g) was immediately processed for DNA extraction, ensuring the integrity of the microbial community. Negative field controls (sterilized water collected using the same protocol and equipment) were also obtained and processed in the same way as field samples to monitor possible contamination.

Figure 1 Distribution and geographical coordinates of the aquatic ecosystems investigated in fragments of Atlantic Forest, Rio de Janeiro State, Brazil.

Red arrows indicate brackish coastal lagoons and blue arrows freshwater environments.

DNA extraction and Illumina library construction

Total DNA extraction was performed using the PowerSoil® DNA Isolation kit (MoBio Laboratories, Carlsbad, CA USA). DNA yields were measured using the Qubit® 2.0 Fluorometer (Thermo Scientific, Waltha, MA, USA). The universal primers 528F (5′-GCG GTA ATT CCA GCT CCA A-3′) and 706R (5′-AAT CCR AGA ATT TCA CCT CT-3′) (Elwood, Olsen & Sogin, 1985; Cheung et al., 2010) were used to amplify the V4 region of the eukaryotic 18S rRNA gene in PCR reactions with the Phusion® High-Fidelity PCR Master Mix (New England Biolabs, USA). The amplicons were sequenced with an Illumina HiSeq 2500 sequencer (Illumina Inc., San Diego, CA, USA), and 2 × 250 bp reads were generated. Raw sequences are available through the project number PRJEB37554 on the European Nucleotide Archive (ENA).

Bioinformatics analyses

Sequencing reads from all samples were first merged with Flash v1.2.11 (Magoč & Salzberg, 2011) and then processed with Quantitative Insights Into Microbial Ecology 2—QIIME2 2022.2 (Bolyen et al., 2019) for demultiplexing and remotion of adaptors, using the q2-demux and q2-cutadapt (Martin, 2011) plugins. The reads were filtered to a minimum Phred quality score of Q20, denoised, dereplicated, and chimerical sequences were eliminated using the q2-quality-filter (Bokulich et al., 2013) and the q2-dada2 plugins (Callahan et al., 2016), respectively. Reads shorter than 210 bp length were also discarded. The amplicon sequence variants were clustered into operational taxonomic units (OTUs) using the q2-vsearch plugin (Rognes et al., 2016) and the open-reference method (Rideout et al., 2014) against the SILVA reference database version 138 (Quast et al., 2012). Sequences with ≥97% similarity were assigned to the same OTU. A sklearn classifier pre-trained on SILVA 138, region 515F/806R, was used to the taxonomic annotation of OTUs (Bokulich et al., 2018) with the q2-feature-classifier plugin (Pedregosa et al., 2011). OTUs from putative multicellular organisms (i.e., assigned to Metazoa, Embryophyta and Fungi) were removed, as well as the ones represented by less than 10 sequences, for noise reduction (Behnke et al., 2011).

Functional assignments of OTUs

The obtained taxonomy table was manually verified and OTUs were assigned to three major functional groups following Singer et al. (2021) as consumers (Ciliophora, Rhizaria, Obazoa non-Ichthyosporea, CRUMs, Amoebozoa, non-Ochrophyta, non-Peronosporomycetes Stramenopiles and Centrohelida), phototrophic (Archaeplastida, Ochrophyta, Prymnesiophyceae and Cryptophyceae) and parasitic (Apicomplexa, Ichtyosporea, Peronosporomycetes, Phytomyxea, Perkinsidae, Syndiniales and Rozellomycota). Since these groups may include organisms with different functional roles, we analyze each OTU classified and consider the least inclusive taxonomic level to assign function (Table S1). Some groups of Chrysophyceae have lost their photosynthetic ability secondarily (Dorrell et al., 2019). Therefore, we considered as consumers those OTUs assigned to Oikomonas, Spumella, Apoikia, Poteriospumella and Paraphysomonas, also following Singer et al. (2021). Other genera were considered phototrophic and the OTUs not classified at this level were tagged with unknown function.

Diversity studies

We estimated the α diversity, i.e., the number of observed OTUs, the Shannon’s index H′ (Whittaker, 1972), and the Simpson’s index D (Simpson, 1949) for each sample with the R-package phyloseq (McMurdie & Holmes, 2013). OTU richness in freshwater and brackish samples was also estimated using species accumulation curves (functions specaccum, R-package vegan v. 2.6–2) (Oksanen et al., 2022). We assessed the similarity patterns among protist communities (β diversity) using principal coordinate analysis (PCoA) based on Bray-Curtis dissimilarities obtained from the composition and relative abundance of sequences. Significance of differences between groups was assessed using the Permanova test (adonis function R-package vegan with 1,000 permutations). We tested for differences between ecosystems for α and β diversity indices by pairwise tests for multiple comparisons of mean rank sums (Nemenyi test, p < 0.05; function NemenyiTest, R package DescTools). We also use this approach to test the differences between functional groups based on the relative abundance of OTUs.

Results

Protist community richness and heterogeneity in freshwater and brackish systems from Atlantic Forest

The sequencing generated a total of 1,742,075 reads. After all quality filtering steps, 253,637 reads with an average sequence length of 350 bp remained for downstream analysis. After clustering at 97% similarity, a total of 2,692 OTUs were retrieved. Subsequently, OTUs not assigned to the phylum taxonomic category, identified as ‘unclassified’, ‘uncultured’ and ‘incertae sedis’ (256 OTUs) were removed, as well as sequences from putative non-protist organisms, as OTUs assigned to Metazoa (383 OTUs), Fungi (408 OTUs) and Embryophyta (55 OTUs). In the end, a total of 1,590 OTU sequences assigned to protist groups were retained (Table S1) and used for the diversity analyses.

OTU richness tended to approach a saturation plateau, as shown by the species accumulation curves (Fig. 2A). Protist richness was significantly higher in freshwater (1,148 OTUs) than in brackish samples (419 OTUs). Only 23 OTUs were shared between these two sampling groups (Fig. 2B). However, when abundance data are considered, the means of the α diversity indices do not differ significantly between freshwater and brackish environments by t-test (Fig. 2C), suggesting that a more even distribution of sampling effort could equalize the OTU richness retrieved from these environments. The samples with the highest OTU richness were from Guapiaçu river (265 OTUs) and from Três Picos Park (272 OTUs). The sample with the highest α diversity value was from the Boa Vista stream (Table S2). All these highly diverse sites are located in the Serra dos Órgãos National Park.

Figure 2 Richness and diversity of protist OTUs.

(A) Species accumulation curves by sample. (B) Venn’s diagram of the total amount of OTUs in freshwater, brackish water and shared by both sample groups. (C) α diversity metrics. Number of unique OTUs; Shannon = Shannon’s index H; Simpson = Simpson’s index D. The average values of the α diversity indexes do not differ significantly between freshwater and brackish sampling groups (p-value > 0.05).

Beta diversity was highest among freshwater samples (0.968 ± 0.06) and significantly lower among brackish water samples (0.911 ± 0.15) (Fig. 3A). Principal Coordinate Analysis revealed that protist communities from brackish and freshwater environments are distinctly structured (Fig. 3B). The adonis test showed that the richness and abundance of the protist OTUs are significantly different between freshwater and brackish samples (p-value < 0.01; Fig. 3B). This dissimilarity between the two environments with respect to the protist communities was also confirmed by the Nemenyi’s test for multiple comparisons (p-value = 0.0014).

Figure 3 Beta diversity measures.

(A) Bray-Curtis distances within each ecosystem based on protist OTU composition (presence-absence data) and relative abundances. (B) Ordination plot (principal coordinates analysis = PCoA) of protists communities based on Bray-Curtis dissimilarities. The protist OTU composition in freshwater and brackish samples differs significantly (p-value < 0.01).

Taxonomic and functional diversity

The 1,590 OTUs were distributed among seven of the protist supergroups (sensu Burki et al., 2020). As expected, most of the sequences were assigned to the clade TSAR (1,292 OTUs), representing more than 80% of the total diversity. Archaeplastida (177 OTUs) followed with 11% of the total OTU diversity. The other groups were much less represented, such as Obazoa (44 OTUs), Amoebozoa (31 OTUs), Cryptista (23 OTUs), CruMs (13 OTUs), and Haptista (9 OTUs) (Table S1).

These OTUs were assigned to 26 major protist phyla (Fig. 4). Ciliophora is the most represented (451 OTUs), followed by Diatomea (336 OTUs), Chlorophyta (161 OTUs) and Cercozoa (153 OTUs), together accounting for over two-thirds of the sequences. The relative abundance in brackish lagoons is dominated by Diatomea, Ciliophora and Dinoflagellata. Other major protist lineages are relatively more abundant or exclusive to freshwater (Fig. 4). Diatomea was the only group with higher OTU richness in brackish water (Fig. 4). The most represented at the genus level were the bacillariophycean diatoms Navicula, Amphora, Pinnularia and Nitzschia with more than 20 OTUs each (Table S1). A single OTU assigned to a marine haptophyte of the genus Isochrysis was detected exclusively in brackish samples (Table S1). From the total, only 418 OTUs were detected in relative abundances ≥1%. This represents 26.3% of the total data set (Table S3). Some of these OTUs showed relative abundances greater than 20% in the samples (Table 1). In particular, ciliates of the genera Paramecium and Laurentiella showed relative abundances of 96% and 80%, respectively, in some freshwater samples. Overall, ciliates are among the top five most abundant protists in the Brazilian Atlantic Forest (Fig. 4; Table S4).

Figure 4 Schematic phylogenetic tree of the main protist lineages, their relative abundances and OTU richness in freshwater (cyan) and coastal (coral) aquatic ecosystems of the Atlantic Forest.

The pie chart represents the relative abundance of reads. Numbers at the right of the pie chart are the total OTU richness of each taxon and the barplots represent the distribution of these OTUs in each ecosystem. Protist groups with highest OTUs richness are indicated numerically. Ciliophora dominates the diversity in freshwater systems while Diatomea is the richest and most abundant group in brackish waters.

Table 1 OTUs whose abundance exceeded 20% in the samples.

Protist main taxa	Identification at genus rank	Environment of highest abundance	Highest abundance (%)	
Ciliophora	Paramecium	Freshwater	0.96	
Ciliophora	Laurentiella	Freshwater	0.80	
Ciliophora	Frontonia	Freshwater	0.65	
Ciliophora	Blepharisma	Freshwater	0.53	
Ciliophora	Zoothamnium	Brackish	0.48	
Labyrinthulomycetes	Labyrinthula	Freshwater	0.46	
Diatomea	Synedra	Freshwater	0.44	
Dinoflagellata	Blixaea	Brackish	0.42	
Phragmoplastophyta	Spirogyra	Freshwater	0.41	
Ciliophora	Heliophrya	Freshwater	0.40	
Diatomea	Amphora	Brackish	0.31	
Diatomea	Pleurosigma	Brackish	0.27	
Ciliophora	Prorodon	Freshwater	0.27	
Diatomea	Gyrosigma	Brackish	0.25	
Diatomea	Stenopterobia	Freshwater	0.22	

We investigate the functional diversity of protists in the two environments, expressed in relative abundance of consumers, phototrophics and parasites (Fig. 5). Of the total OTUs, 848 were attributed to consumers (51.5%), 602 to phototrophics (43.3%), 103 to parasites (5.25%), and 37 OTUs (5.5%) were assigned to groups of organisms that can functionally range from phototrophs to heterotrophs, so we cannot unambiguously assign their functional roles (Table S1). For statistical and graphical purposes, we considered only the 418 OTUs ≥1% abundant in the functional profile analyses (Table S3). Our results showed a remarkable functional homogeneity between the two ecosystems, with non-significant differences between them according to the Nemenyi test (p-value > 0.05). Consumers dominate the richness in freshwater, corresponding to more than 50% of the OTUs in this environment, while in brackish water there is a higher richness of phototrophic protists (Table 2), although the relative abundance of functional groups was statistically equivalent in both environments (Fig. 5).

Figure 5 Relative abundance of OTUs assigned to consumers, parasitic or phototrophic protists in freshwater (cyan) and brackish (coral) aquatic systems of the Brazilian Atlantic Forest.

OTUs representing groups of organisms that can functionally range from phototrophs to heterotrophs are indicated as “NA” (not assigned). Relative abundances do not differ statistically by functional group of protists in these ecosystems (Nemenyi test p-value > 0.05).

Table 2 Distribution of the functional diversity of protists in the Brazilian Atlantic Forest.

Functional group	Ecosystem	Number of OTUs	Corresponding %	
Consumer	Freshwater	172	41.1	
Consumer	Brackish	26	6.2	
Parasites	Freshwater	27	6.4	
Parasites	Brackish	1	1.0	
Phototrophics	Freshwater	111	26.5	
Phototrophics	Brackish	68	16.2	
NA	Freshwater	13	3.1	
NA	Brackish	0	0	

Discussion

Protist communities in coastal lagoons and freshwater systems of the Brazilian Atlantic Forest are equally diverse

The vast majority of biodiversity studies using HTS technology have been conducted in marine environments (e.g., Rychert et al., 2014; de Vargas et al., 2015; Massana et al., 2015; Gimmler et al., 2016). Relatively few metabarcoding surveys have been dedicated to investigating the diversity of inland waters, which are potentially much more diverse (e.g., Zinger, Gobet & Pommier, 2012; Balzano, Abs & Leterme, 2015; Fernandes et al., 2021). In understudied geographic regions, such as South America, these approaches are even rarer. We investigate for the first time the taxonomic and functional diversity of major protist lineages in freshwater and brackish systems located in fragments of the Brazilian Atlantic Forest. Specifically, the brackish systems studied are coastal lagoons, located in densely populated areas and considered one of the most impacted environments in the world (Esteves et al., 2008).

Our results showed that the diversity of protists in these coastal lagoons does not significantly differ from that in freshwater in terms of OTU richness and relative abundances, even though the number of samples analyzed from coastal lagoons is much smaller (six brackish vs 17 freshwater samples). This result is in contrast to Remane’s concept of a minimum number of species in transitional waters (Remane, 1934), which argues that taxonomic diversity is lowest at salinities between 5 and 8 psu (Kinne, 1971). However, this concept has been shown to be based on insufficient knowledge of the taxonomic composition of organisms (Telesh et al., 2011; Telesh, Schubert & Skarlato, 2011). Conversely, bacterial and protist diversity is usually higher in brackish waters (Telesh, Schubert & Skarlato, 2013; Santoferrara, Rubin & McManus, 2018) or comparable to other environments (e.g., Hu et al., 2016). Other diversity surveys in Brazilian coastal lagoons have reported high zooplankton diversity (Reid & Esteves, 1984; Branco, de Assis Esteves & Kozlowsky-Suzuki, 2000), with α diversity indexes comparable to that of Amazonian lakes (Carneiro, Bozelli & Esteves, 2003; Esteves et al., 2008). Here, we have observed the same pattern for protists.

Although our findings indicate that freshwater and brackish systems from Atlantic Forest are similar in terms of protist OTU richness and structure, including in relation to the functional profile of the organisms (details below), these two ecosystems differ significantly in terms of OTU taxonomic composition. Most OTUs were detected in either freshwater or brackish water, so the protist community composition differed significantly between the two environments (Fig. 3B). This was expected, as they are completely different ecosystems, and as previously reported for ciliates (Fernandes et al., 2021). However, a total of 23 OTUs were recorded in both ecosystems, including the bacillariophycean diatoms Navicula, Amphora and Gomphonema, and the ciliates Paramecium and Laurentiella, which can tolerate a wide range of salinity levels (Wilson, Cumming & Smol, 1996; Clavero et al., 2000; Smurov & Fokin, 2001).

Bray-Curtis distances were significantly greater among freshwater samples. This indicates greater heterogeneity within this sampling group in terms of OTU composition and abundance compared to brackish samples. This result was expected because the freshwater samples analyzed were taken from different water bodies, i.e., ponds, rivers, streams and waterfalls, which represent totally different environments, with different flow conditions, oxygen levels, etc. On the other hand, brackish coastal lagoons tend to be more similar to each other than to continental or marine waters, due to shared features such as strong physico-chemical gradients with adjacent ecosystems, variations in salinity and shallowness, among others (Pérez-Ruzafa et al., 2011), especially if they are geographically close and connected. Thus, these environments may share a basic set of species adapted to the same environmental conditions, or ecological guilds (Pérez-Ruzafa et al., 2011). However, due to the reduced number of brackish samples analyzed, this pattern should be considered with caution.

The freshwater samples located at Serra dos Órgãos National Park, a federal protected area (Rylands & Brandon, 2005), were the richest in protist OTUs and with the highest α diversity indexes in general. The potential of Brazilian protected areas for the discovery of new protistan taxa is underlined by the number of unclassified OTUs beyond class rank in these samples (Table S1). Indeed, several new protist taxa have recently been described from the same sampling ecosystems here investigated (e.g., Paiva et al., 2016; Campello-Nunes et al., 2015; Campello-Nunes et al., 2020; Campello-Nunes et al., 2022). This also emphasizes the importance of expanding sampling efforts in neotropical environments to enhance our comprehension of the global protist diversity.

Functional groups are homogeneously represented in freshwater systems of the Brazilian Atlantic Forest

Ciliates have been the richest and relatively most abundant group in the studied freshwater ecosystems. These heterotrophic organisms have a wide range of life styles and have been successful in the colonization of diverse environments (Lynn, 2008). In fact, it is one of the most represented protist groups not only in the Atlantic Forest (Simão et al., 2017; Fernandes et al., 2021), but also in other Brazilian biomes (de Araujo et al., 2018; Lentendu et al., 2019). A previous study suggested that nearly one third of the ciliate OTUs share less than 97% sequence identity with reference sequences and may represent new ciliate taxa or nominal morphotypes that have already been described, but for which 18S rRNA gene sequences have not yet been deposited in reference databases (Fernandes et al., 2021). However, heterogeneity in rRNA copy numbers in ciliate macronuclei may overestimate their relative abundances (Gong et al., 2013; Geisen et al., 2015). The second most represented group overall and the only group with higher OTU richness and relative abundance in brackish water was Diatomea, mostly the photosynthetic Bacillariophyceae, also following previous surveys in estuaries and coastal lagoons (Roselli et al., 2013; Carstensen, Klais & Cloern, 2015; Leruste et al., 2019; Stefanidou et al., 2020). This success can be attributed to the ability of these organisms to adapt to the severe environmental fluctuations inherent to transitional environments (Snoeijs & Weckström, 2010).

Regarding the functional profile of protist communities, we detected a remarkable functional homogeneity between freshwater and brackish ecosystems, with non-significant differences between them in terms of relative abundances (Fig. 5). This means that there is no dominance of a specific functional group, with the proportions of consumers, phototrophics and parasites roughly balanced in the investigated freshwater environments. The same applies for the investigated brackish systems, in which the proportions of heterotrophic and phototrophic protists are equivalent. However, only a single OTU classified as an apicomplexan parasite was detected (Table S4), revealing low richness and abundance of protist parasites in brackish environments of the Atlantic Forest. Apicomplexa is an extremely diverse group and usually occur in high abundances in a variety of environments, including soils, most commonly infecting metazoans (Geisen et al., 2015; Mahé et al., 2017).

Heterotrophs protists contributed more to freshwater richness than phototrophs, contrary to previous studies (e.g., Singer et al., 2021; Garner et al., 2022). In marine waters there is also a predominance of consumers, as detected by the TARA Oceans expedition (de Vargas et al., 2015). In fact, heterotrophic protists act as primary consumers, transferring significant amounts of bacterial production to higher trophic levels, contributing to nutrient cycling in aquatic food webs (Azam et al., 1983), therefore are essential components of planktonic communities in aquatic systems in general (Nagata, 1986; Jürgens & Massana, 2008). However, we detected a higher richness of phototrophic protists in brackish systems compared to other trophic groups, suggesting a protagonist of microbial photosynthesis in this ecosystem. The functional roles of protists have been extensively studied in marine waters (e.g., Caron et al., 2012, 2017), and comparatively less investigated in continental environments, such as soils and freshwater (Geisen et al., 2018; Singer et al., 2021). Investigating the taxonomic and functional diversity of protists is essential to better understand the evolution, geographic distribution patterns, and ecological roles of these organisms in the Neotropics (Ritter et al., 2021), besides being the starting point for the development of public policies for sustainability and environmental protection. Overall, our study provides valuable information on the taxonomic and trophic profile of the protist communities from the freshwater and coastal brackish systems of the Brazilian Atlantic Forest.

Supplemental Information

Supplemental Information 1 Protist OTU Ids, their taxonomic and functional assignments.

Click here for additional data file.

Supplemental Information 2 Alpha-diversity metrics by sample.

Observed = number of unique OTUs; Shannon = Shannon’s index H; Simpson = Simpson’s index D.

Click here for additional data file.

Supplemental Information 3 The OTUs that occur at above 1% of relative abundance in the investigated environments and their functional roles.

Click here for additional data file.

Supplemental Information 4 Richness, relative and absolute abundances of the main protist lineages in freshwater and brackish systems of the Brazilian Atlantic Forest.

Click here for additional data file.

We thank the team of the Laboratory of Protozoology of the Universidade Federal do Rio de Janeiro for the technical support.

Additional Information and Declarations

Competing Interests

Author Contributions

Data Availability

The authors declare that they have no competing interests.

Vanessa Carvalho da Silva analyzed the data, prepared figures and/or tables, authored or reviewed drafts of the article, and approved the final draft.

Noemi Fernandes conceived and designed the experiments, performed the experiments, analyzed the data, authored or reviewed drafts of the article, and approved the final draft.

The following information was supplied regarding data availability:

The data is available at European Nucleotide Archive (ENA): PRJEB37554.

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
