# Peer review of "Protist taxonomic and functional diversity in aquatic ecosystems of the Brazilian Atlantic Forest"

_PeerJ, doi:10.7717/peerj.15762_

## Round 0.1 · original submission · Major Revisions

Dear authors,

After careful analysis by three experts in the field, your manuscript provides interesting insight and is overall well-written and the criticism provide is positive and insightful. I kindly request that you address all the points that the reviewers are recommending to revise. I personally recommend toning down the manuscript since the sample size is small and please provide more detail in the text regarding the analysis done for assessing diversity.

Thank you for considering PeerJ for publishing your work.

With best regards,
Bernardo

Reviewer 1 ·

Basic reporting

no comment

Experimental design

reduced sampling size (see below)

Validity of the findings

Given the relatively limited sampling size (especially in brackish lagoons), the results obtained need to be taken with caution; the text should be more nuanced.

Additional comments

This paper describes the taxonomical and functional diversity of planktonic protists in two ecosystem types found in the Mata Atlantica region in Brazil, freshwater streams and coastal brackish lagoons. Diversity was assessed by metabarcoding the v4 region of the gene coding for the 18S rRNA. Authors find, surprisingly, a comparable diversity in brackish and freshwater environments, but a higher beta diversity in freshwater. Also, they find a functional homogeneity between both environment types. Freshwater was dominated by ciliates while brackish environments had more diatoms.
Overall, I think it is a good paper that should be accepted for publication. However, given the relatively limited sampling size (especially in brackish lagoons), the results obtained need to be taken with caution; the text should be more nuanced.
Here are some suggestions that should, I think, improve the article
Line 23: It's hypervariable, with y. By the way, although this term appears often in the literature, I think that no region of the 18S can be really qualified as hypervariable, as this gene is slow evolving in most eukaryotic taxa
Line 48: The term symbionts includes parasites and mutualists as well as commensals. I would write symbionts (i.e. parasites, parasitoids, mutualists and commensals)

Line 65: Protists are also often planktonic... please clarify this sentence
Lines 87-91: Last paragraph of the introduction: Here, I would emphasize the importance of the comparison between brackish and freshwater environments, which leads to th ekey results of this article

Line 99: Was there any prefiltration step to remove large particles? If so, what was the mesh size?

Line 114: Here I would specify 2x250 bp reads

Line 130: LKM15 belongs to the Rozellomycota, a group of unicellular parasites related to Fungi. I think that it makes most sense to include them in the study

Line 138: Most MAST are not photosynthetic.

Line 145: Genus names should be in italics

Line 198: represented. Please change according in the rest of the text

Line 202: Chlorophyta

Lin 210: I would refrain from mentioning species here. 18S is a conserved marker and does not have the resolution power to discriminate between species, and even less when a 3% threshold is applied.

Line 237: “have been conducted in marine environments"

Line 283: I would love to see that local environmental policies enhance microbial biodiversity. However, I think that this is clearly an overstatement. Dtermining the factors that influence protist diversity is one of the main goals in protistology, which has not been reached yet. What policies are referred to here? Do they affect all groups at the same level? Etc... In summary, I would just remove that sentence!

Line 286: This is mostly due to the fact that Neotropics have been undersampled. Similar results can be found in the Paraná floodplain in Argentina, which is not protected:
Metz S. et al. (2022) Freshwater protists: unveiling the unexplored in a large floodplain system. Environmental Microbiol. 24: 1731 – 1745

Line 287: Again, this is due to limited knowledge of the ecosystems, which are understudied not only molecularly but also by classical approaches.

Line 297: Relative abundances of the less-represented taxa depends also on the most represented ones. If Paramecium can represent up to 94% of all sequences (probably due to a bloom), then almost all other taxa will fall into the "rare biosphere". Moreover, some "rare" sequences are infrequent because there are biased against during PCR amplification. Then, a part of this rare biosphere is constituted by artefactual sequences, tag jumping etc. Altogether, I suggest that authors should be very careful when discussing the existence of a rare biosphere, or maybe avoid this discussion.

Line 336: Here, the authors should cite the seminal paper from Azam et al., 1983 ("microbial loop") or even Pomeroy et al., 1974

Reviewer 2 ·

Basic reporting

The paper from Carvalho da Silva and Fernandes is well put together, competently made, and constitutes an interesting contribution in the field.
The language might benefit from more polishing and the feedback from a fluent English speaker. Everything is generally clear, but there are a lot of typos and awkward sentences. I noted down some below, but it is not an extensive list.
I have only one major criticism and several minor ones, but I otherwise recommend publication.


MAIN CONCERNS

While the methodology is overall standard and done correctly, I have two issues, both related to lines 126-132.
The first issue is probably just an omission, but it seems like all the details concerning the taxonomic assignment methods are missing. These should absolutely be provided, since they are key to the paper.
The second issue concerns the method used to define the OTUs. This is clearly stated by the authors, but it sounds really dated to me – why did they use similarity clustering, at a very low threshold of 97% at that, instead of just obtaining ASVs? Similarity clustering is a very coarse and outdated approach, and the authors should provide a reason for using it. Nevertheless, most of the paper deals with very high-level taxonomy, so it wouldn’t be a huge deal in terms of results, EXCEPT in the part where species identification is mentioned (lines 210-212). If the authors stick to a 97% threshold, that part must be removed – it is absolutely impossible to talk about species level for 97%-similarity OTUs. You are almost certain to have multiple species, if not genera, in each.


MINOR COMMENTS & TYPOS

Line 23, typo, “hipervariable”.
Line 28, “freshwaters” (should be “freshwater”)
Line 48, “[…]parasites or symbionts” – parasites ARE symbionts…
Line 76, “richest in species diversity” – since the authors talk throughout the paper about the massive unexplored species diversity represented by microbes, it might be a good idea to be more specific on what is referred to here
Line 100, “[…]extraction in les […]” – “in” is unnecessary
Lines 104-105. Shouldn’t the outputs of the controls be reported in the results?
Line 112 and elsewhere. “rDNA” is a bad piece of nomenclature anyway, but it is used especially incorrectly here (“rDNA gene”). Please stick consistently to “18S [or SSU] rRNA gene”
Line 114, “Hiseq” should be “HiSeq”
Lines 135-140. I consider functional assignment based on broad SSU taxonomy very unreliable, but I might be in the minority, and the authors cite their sources. However, some things seem a bit too generalized here. What about chlorarachniophytes in Rhizaria? Why are MASTs considered all phototrophic? What about parasitic ciliate clades, like Astomatia, Apostomatia and (most) Scuticociliatia?
Line 139 – “Parasitic” shouldn’t be capitalized
Line 145 – genera should be in italics
Lines 175-177 – this sentence is far too strong. Saturation curves prove that additional sequencing, everything else being equal, would not have provided additional results, but there are swathes of potential systemic errors in these projects, that should make everyone hesitant to claim that “the total diversity was completely recovered from the samples”
Lines 186-187 – are you sure this is also not due to the very different number of samples? From the graph, it looks like the few low-diversity samples simply have a disproportionate effect on the brackish average (see also lines 270-272)
Line 198 and elsewhere: “represented”, not “representative”
Line 202, typo, “Clorophyta”
Line 205, “respectively”. Unclear what it refers to
Line 221 and elsewhere – when used as a noun it should be “phototrophs”, not “phototrophics”
Lines 325-329 – is it possible you simply did not collect apicomplexan hosts?
Lines 337-338 – sentence unclear
Line 343 – “besides to”. The “to” is unnecessary

Figure 2 – “Observed” is a weird term taken straight from QIIME2. Please change it to something more clear (eg “OTU richness” or “OUT number”)
Figure 4 – there is no number to the left of pie charts

Experimental design

No comment (please see above)

Validity of the findings

No comment (please see above)

Additional comments

No comment (please see above)

·

Basic reporting

The English language is clear and unambiguous written. However, parenthesis are missing in each in Line 152 and 153.
Your introduction is good. Nevertheless, if there are I suggest that you use updated data regarding references in lines 76-79. In Lines 97 and 119 the year of the citation is wrong (Fernandes et al., 2020), it is 2021.
The structure of the article is according to the Instructions for authors.
In the Figure 4 (Lines 209 and 210) is not clear that the genus Isochrysis was exclusive for brackish water.

Experimental design

The submission clearly defines the research question, which is both relevant and meaningful. This text highlights the identification of the knowledge gap and the contribution of the study in addressing that gap.
It is original primary research, and the experimental design was well proposed. The methods section provides enough information to allow reproducibility. However, I have a question regarding the Sampling: were the water and resuspended sediment collected from different depth? Could it have any repercussions on the interpretation of data?

Validity of the findings

It seems that the research question in this article arises from a previous publication of the authors (Fernandes et al., 2021). In this case, the rationale and benefit are clearly stated, and the results are innovative.
All underlying data exhibit robustness, statistical soundness, and effective control.
The conclusions are clearly articulated, linked to the original research question, and limited to supporting the obtained results.

Additional comments

Could the authors suggest which is the ecological and biological relevance of the distribution of the functional diversity shown in Table 2?

---

## Round 0.2 · accepted · Accept

Dear authors,
I am happy to announce that the experts that reviewed the manuscript who commented on some issues that needed to be attended to have recommended the manuscript for publication. I agree that the study is well-designed and presented. In the proof stage please ensure that your manuscript is correctly formatted in the proof stage.

Congratulations!
Best regards and all the best for your research moving forward.
Kindly,
Bernardo

Reviewer 1 ·

Basic reporting

The quality of the language used is good, and the references added in the second version are now more complete. Just one comment: many of the new citations have a quite different format than the others. Authors should check this carefully (I guess this can be done during the proofreading step anyway).

Experimental design

I appreciate that the conclusions on the brackish water communities have been toned down given the little number of samples. Also, OTUs are not assigned to species anymore, which is a good thing.

Validity of the findings

These points were already ok in the first version I think

Reviewer 2 ·

Basic reporting

No comment.

Experimental design

No comment.

Validity of the findings

No comment.

Additional comments

No comment.